# Single-Cell Sequencing Reveals the Regulatory Role of Maresin1 on Neutrophils during Septic Lung Injury

**DOI:** 10.3390/cells11233733

**Published:** 2022-11-23

**Authors:** Fuquan Wang, Ming Chen, Chenchen Wang, Haifa Xia, Dingyu Zhang, Shanglong Yao

**Affiliations:** 1Department of Anesthesiology, Union Hospital, Tongji Medical College, Huazhong University of Science and Technology, Wuhan 430022, China; 2Department of Anesthesiology, Institute of Anesthesia and Critical Care Medicine, Union Hospital, Tongji Medical College, Huazhong University of Science and Technology, Wuhan 430022, China; 3Wuhan Jinyintan Hospital, Wuhan, 430023, China

**Keywords:** sepsis-induced lung injury, Maresin1, neutrophils, cellular heterogeneity

## Abstract

Acute lung injury (ALI) is the most common type of organ injury in sepsis, with high morbidity and mortality. Sepsis is characterized by an inappropriate inflammatory response while neutrophils exert an important role in the excessive inflammatory response. The discovery of specialized pro-resolving mediators (SPMs) provides a new direction for the treatment of a series of inflammatory-related diseases including sepsis. Among them, the regulation of Maresin1 on immune cells was widely demonstrated. However, current research on the regulatory effects of Maresin1 on immune cells has remained at the level of certain cell types. Under inflammatory conditions, the immune environment is complex and immune cells exhibit obvious heterogeneity. Neutrophils play a key role in the occurrence and development of septic lung injury. Whether there is a subpopulation bias in the regulation of neutrophils by Maresin1 has not been elucidated. Therefore, with the well-established cecal ligation and puncture (CLP) model and single-cell sequencing technology, our study reveals for the first time the regulatory mechanism of Maresin1 on neutrophils at the single-cell level. Our study suggested that Maresin1 can significantly reduce neutrophil infiltration in septic lung injury and that this regulatory effect is more concentrated in the Neutrophil-Cxcl3 subpopulation. Maresin1 can significantly reduce the infiltration of the Neutrophil-Cxcl3 subpopulation and inhibit the expression of related inflammatory genes and key transcription factors in the Neutrophil-Cxcl3 subpopulation. Our study provided new possibilities for specific modulation of neutrophil function in septic lung injury.

## 1. Introduction

Sepsis is life-threatening multiple organ failure resulting from a dysregulated host response to infection, posing a serious worldwide health burden [1]. Among the damaged organs, the lung is the most vulnerable organ during sepsis [2] and sepsis-induced acute lung injury (ALI) is one of the most critical prognostic factors for mortality in sepsis patients [3].

Despite extensive research that was performed on sepsis-induced ALI, current treatment strategies are still dominated by supportive ventilation strategies. The pathophysiological mechanisms associated with the development and therapeutic targets of sepsis-induced ALI have so far not been definitively elucidated [4].

The inappropriate inflammatory response is an important pathological feature of septic lung injury. The resolution of inflammation was once thought to be a passive effect due to the reduction of pro-inflammatory mediators, while the discovery of the specialized pro-resolving mediators (SPMs) suggested that this process is a well-planned active and coordinated active process [5,6]. As a specific pro-inflammatory mediator derived from docosahexaenoic acid (DHA) in macrophages [7], Maresin1 was demonstrated to have extensive regulatory effects on immune cells in inflammatory-related diseases including sepsis [8]. We also performed some exploration on the role of Maresin1 during sepsis and our findings suggest that Maresin1 can promote M2 macrophage polarization by promoting peroxisome proliferator-activated receptor-gamma activation [9]. In addition, Maresin1 could reduce neutrophil accumulation and overcome Lipopolysaccharide-mediated inhibition of neutrophil apoptosis, etc. [10,11].

Neutrophils have always been an important focus for exploring and solving the problem of septic lung injury. As an important component of the innate immune system, the overactivation of neutrophils is one of the important pathological mechanisms of septic lung injury [12]. Neutrophils can induce organ damage through a variety of pathways, including the release of inflammatory mediators such as cytokines and reactive oxygen species. The neutrophil infiltration was considered to be the rate-limiting step in the progression of ALI [13]. Neutrophils are heterogeneous [14,15,16], especially in different environments such as homeostasis and bacterial infection, neutrophils show obvious heterogeneity [17]. Exploring the heterogeneity of neutrophils in septic lung injury can help us discover whether targeting specific neutrophil subsets is a more effective treatment modality. However, current studies of neutrophils in septic lung injury have not distinguished their heterogeneity.

Our previous studies have shown that Maresin1 has an effective protective effect on septic lung injury [11,18], which can significantly improve lung function and reduce the inflammatory response of lung tissue. However, it is unknown whether Maresin1 acts consistently on all subpopulations of neutrophils under inflammatory conditions. Single-cell RNA sequencing (scRNA-seq) is a powerful tool for exploring cellular heterogeneity [19]. Therefore, by establishing the most commonly used sepsis model of cecal ligation and puncture (CLP), we used single-cell sequencing technology to reveal the regulatory mechanism of Maresin1 on neutrophils in septic lung injury. Our studies aimed to investigate the mechanism of Maresin1 regulation of neutrophils at the single-cell level in septic lung injury and to uncover the underlying cellular pathophysiologic mechanism.

## 2. Materials and Methods

### 2.1. Ethics Statement

All animal experiments were approved by the Institutional Animal Care and Use Committee of Huazhong University of Science and Technology (IACUC Number: 2629).

### 2.2. Animals

Twenty-one C57BL/6 male mice (six weeks old, 20–25 g), obtained from Beijing Weitong Lihua Experimental Animal Technical Co., Ltd., were randomly divided into 3 groups, including the Sham group, the CLP group, and the Maresin1 group. All mice were acclimated for one week before starting the experiment. The laboratory conditions were: a 12 h light–dark cycle, a humidity of 40–60%, and a temperature of 22–24 °C. All the mice were allowed free access to water and food.

The construction of the CLP model is as we described in our previous studies [20]. Briefly, we first anesthetized mice with 2% sodium pentobarbital at a dose of 80 mg/kg. We made a 1.5 cm incision along the midline of the mouse abdomen, took out the cecum, ligated the middle and perforated with a 20-gauge needle. After squeezing a little feces, the cecum was restored to the abdominal cavity, and the abdominal incision was sutured layer by layer. The mice in the Maresin1 group were administered 1 ng of Maresin1 (Cayman Chemical, Ann Arbor, MI, USA; Cat.10878) in 100 μL of normal saline via the tail vein. Our previous studies [11,20] showed that 1ng of Maresin1 can exert an effective protective effect on mice with septic lung injury, therefore we choose 1 ng as the experimental dose. After 24 h, the abdominal aorta of the mice was cut and the mice were euthanized and the mouse specimens were collected for subsequent experiments.

### 2.3. The Lung Histological Examination

The harvested lung tissue was fixed with 4% paraformaldehyde, embedded in paraffin, and cut into 4 μm sections. Then, the sections were deparaffinized and stained with hematoxylin and eosin (HE) (n = 7 per group). The sections were observed under the ordinary optical microscope and damage scores were calculated. The scoring method was based on the lung injury scoring criteria published by the American Thoracic Society [21].

### 2.4. The Neutrophil Count in Bronchoalveolar Lavage Fluid (BALF)

The right lung of the mouse was ligated and BALF was collected by lavage of the left lung with 1.5 mL of phosphate-buffered saline (PBS) (0.5 mL each). After BALF was centrifuged at 1000 rpm for 10 min, the cell pellet was resuspended with PBS and then stained with Giemsa stain (n = 7 per group). The number of neutrophils was counted under an ordinary optical microscope.

### 2.5. Immunofluorescence Assay

The sections (4 μm-thick, n = 6 per group) were incubated with Ly6G antibody (Thermo Fisher Scientific, Waltham, MA, USA, 14-5931-81) (1:200) and the ICAM1 antibody (Abcam, Boston, MA, USA, ab222736) (1:100) followed by the fluorescently labeled secondary antibod (Wuhan baiqiandu Biotechnology Co., Ltd., Wuhan, China, 1:200, B1000804, B1000805). The positive cells were then observed under the Olympus fluorescence microscope. We observed five fields for each section and took the average value of the number of positive cells for statistics.

### 2.6. Single-Cell Sequenceing

The three Sham group specimens, the three CLP group specimens, and the three Maresin1 group specimens were mixed, respectively, for single-cell RNA sequencing. Single-cell suspensions of lung tissue specimens were prepared according to OE Biotech Co., Ltd.’s advice [22]. The cells were then sorted using Miltenyi mouse CD45 magnetic beads according to the manufacturer’s recommendations and sequenced at a 1:1 ratio. According to the manufacturer’s guidelines, the single-cell libraries were constructed by 10× Genomics Chromium Single Cell 3ʹ Reagent Kits v3.1. The Illumina Nova 6000 PE150 platform was used to sequence the libraries.

### 2.7. Single-Cell Data Preprocessing

Cell Ranger v5.0.0 was used for the initial processing of the scRNA-seq data. We then performed the downstream bioinformatic analyses with Seurat v. 4.1.2 in the R statistical environment. The low-quality cells were removed if they expressed <2000 or >6000 genes/cell, UMI counts <1000 or mitochondrial UMI counts greater than 10% mitochondrial. The “Harmony” package was used to remove batch effects between samples [23]. The cells after quality control were grouped with the “FindNeighbors” and “FindClusters” functions with a resolution of 1. The clusters were then visualized after “RunTSNE” or “RunUMAP”. The “FindMarkers” function in Seurat was used to identify DEGs. Only the genes that were expressed in more than 25% of the cells of a cluster and had an average log (fold change) value greater than 0.5 could be selected as DEGs. We annotated the cell type of each cluster by the expression of canonical markers found in DEGs. The cells expressing two or more canonical markers were considered double cells and also were eliminated.

The final total was 11,025 immune cells, the Sham group was 2992, the CLP group was 4265, and the Maresin1 group was 3768.

### 2.8. Pathway Enrichment Analysis

The gene ontology (GO) enrichment analysis [24] and Kyoto Encyclopedia of Genes and Genomes (KEGG) analysis [25] were performed with the “clusterProfiler” R package [26].

### 2.9. The Gene Regulatory Network Analysis

The transcription factors were assessed by pyscenic [27] analysis on all neutrophils and differentially expressed transcription factors were calculated using the Limma [28] software package.

### 2.10. Statistical Analysis

The validation data were obtained from the public functional genomics data repository of Gene Expression Omnibus (GSE151974) [29]. The statistical analyses and graph generations were performed in R (v 4.1.2) and GraphPad Prism (v 9.0).

## 3. Results

### 3.1. Maresin1 Attenuates the Extent of Sepsis-Induced Lung Damage

As shown in Figure 1A, compared with the Sham group, the mice in the CLP group had disordered lung tissue structure, thickened alveolar septa, accumulation of inflammatory cells in the alveoli, and the pulmonary septum, and bleeding and protein debris exudation in the alveolar cavity. The degree of lung injury of the mice in the Maresin1 group was significantly reduced. The lung damage score was consistent with the pathological changes (Figure 1B).

### 3.2. Maresin1 Reduces Neutrophil Numbers in Bronchoalveolar Lavage

As shown in Figure 1C, the number of neutrophils in the BALF of the CLP group was significantly higher than that of the Sham group (*p* < 0.01); compared with the CLP group, Maresin1 significantly decreased the number of neutrophils (*p* < 0.01) but was still higher than that of the Sham group (*p* < 0.01). The immunofluorescence results (Figure 1D) suggested that the recruitment and infiltration of the neutrophils in the lung tissue of the CLP group were significantly higher than in the Sham group, while Maresin1 could decrease the neutrophil infiltration. In order to observe the distribution of neutrophils more clearly, we performed fluorescent double staining of Ly6g and Icam1 and the results showed that neutrophils in septic lung injury were diffusely distributed (Appendix A).

### 3.3. Maresin1 Alters Neutrophil Composition in Septic Lung Injury

We use single-cell sequencing to reveal the immune microenvironment of lung tissue in septic lung injury. Based on the single-cell sequencing results, we first divided all immune cells in lung tissue into nine main types (Figure 2A) according to the expression of classical markers (Figure 2B), including: “Cycle cell”, “Alveolar Macrophages”, “Mono”, “Macrophages”, “DC cell”, “Neutrophil”, “B cell”, “Nk cell”, “T cell”. Among them, the Cycle cell subpopulation highly expresses Mki67 and Top2A [30] and is a mixed population of immune cells in proliferating state. The top five DEGs of different clusters are shown in Figure 2C. Furthermore, the whole DEGs of different immune cell clusters are listed in Appendix A. As shown in Figure 2A, neutrophils make up a large proportion of the immune cell population, therefore we further divided the neutrophils into four subpopulations (Figure 2C) based on the marker genes (Figure 2D): Neutrophil0-Ccl6, Neutrophil1-Krt83, Neutrophil2-Csta2, Neutrophil3-Cxcl3. Notably, we found that the Neutrophil1-Krt83 subpopulation highly expressed Krt83, which is recognized as the basic hair keratin gene [31]. Therefore, we next used GSE151974 for verification and we found that between the two subpopulations of neutrophils in GSE151974, there was also a subpopulation that highly expressed Krt83 (Appendix A). Compared with normal mice, sepsis changed the proportion of all neutrophil subpopulations. The neutrophil fraction of the Maresin1 group, except for the Neutrophil3-Cxcl3 subgroup, was similar to that of the CLP group (Figure 2E). Therefore, we selected the Neutrophil3-Cxcl3 subpopulation for further analysis. The top 20 DEGs of the Neutrophil3-Cxcl3 subpopulation are shown in Figure 3A. To further confirm the Neutrophil3-Cxcl3 subpopulation, we also performed validation by using GSE151974. There are also neutrophils that express the Cxcl3 gene significantly in GSE151974 (Appendix A). Functional analysis of the DEGs of the Neutrophil3-Cxcl3 subpopulation was subsequently performed. The GO analyses (Figure 3B) suggested that the enrichment of DEGs was mainly involved in the Chemotaxis and recruitment of inflammatory cells, the release of cytokines, etc. The KEGG analyses (Figure 3C) suggested that the enrichment of DEGs was mainly involved in the canonical pathways of neutrophils, including NF-kappa B signaling pathway, Cytokine–cytokine receptor interaction, and the Toll-like receptor signaling pathway. The functional analysis indicated that the Neutrophil3-Cxcl3 subpopulation was closely related to the excessive inflammatory response in septic lung injury.

### 3.4. Maresin1 Inhibits the Expression of Key Genes in the Neutrophil-Cxcl3 Subpopulation

Cxcl3, Ccl3, Ccl4, Icam1, and Il1rn are the main genes involved in the Go enrichment and KEGG pathway enrichment of the Neutrophil3-Cxcl3 subpopulation. We extracted the Neutrophil3-Cxcl3 subpopulation from the total neutrophils for further analysis according to the source of the group. As shown in Figure 4A, the expression of the Cxcl3, Ccl3, Ccl4, Icam1, and Il1rn of the Neutrophil3-Cxcl3 subpopulation was higher in the CLP group than in the Sham group, while Maresin1 intervention decreased the expression of these genes.

### 3.5. Maresin1 Represses Transcription Factors in Neutrophil-Cxcl3

Transcription factors play a crucial role in cell differentiation and fate. Based on the results of pyscenic analysis and the previously published literature, we studied the transcription factors of neutrophils. Subpopulation analysis revealed that Irf5, Junb, Nfkb2, Relb, and Rfx2 transcription factors play important roles in the Neutrophil3-Cxcl3 subpopulation (Figure 4B). Upon further analysis of the Neutrophil3-Cxcl3 subpopulation derived from different groups, we found that Maresin1 could significantly reduce the expression levels of these transcription factors (Figure 4C).

## 4. Discussion

Sepsis is a severe infectious complication characterized by a dysregulated immune response and subsequent multiple organ dysfunction [32]. Although the exploration of the pathological mechanisms and therapeutic targets of septic lung injury has been ongoing in the past few decades, there is still no definite and effective treatment. Sepsis included the initial phase of hyper-inflammatory response followed by a phase of immunosuppression [33]. When an acute inflammatory injury occurs, neutrophils can migrate and aggregate to the foci of the inflammatory reaction infection and play the role of capturing and phagocytizing microorganisms by releasing reactive oxygen species or through neutrophil extracellular nets (Nets). However, hyperactivated neutrophils are also an important factor in promoting the development and exacerbation of acute lung injury [34]. Neutrophils are an important target for the treatment of acute lung injury in sepsis [35].

The discovery of SPMs provides new possibilities for the treatment of inflammatory-related diseases; however, the current regulation of immune cells by SPMs is not precise enough. As an important member of the SPMs family, Maresin1 has shown a positive prospect in the treatment of inflammatory-related diseases. On the one hand, Maresin1 has a positive regulatory effect on a variety of inflammatory-related diseases. On the other hand, Maresin1 has regulatory effects on both innate and adaptive immune cells. However, the regulatory effect of Maresin1 on immune cells is still at the level of specific cell types and there is a lack of research on specific cell subclusters. Whether the regulatory effect of Maresin1 on immune cells is subpopulation-biased has not been demonstrated.

Neutrophils are the most abundant leukocytes in mammals and play a key role in the pathogenesis of sepsis-induced lung injury [36]. However, there is considerable debate about the impact of therapy-induced neutrophil depletion in sepsis, as the impact on bacterial clearance and response to systemic inflammation is unclear [37]. Another important explanation may be the obvious heterogeneity in the number ratio, functional characteristics, transcription factor expression, etc. of different subpopulations of neutrophils under inflammatory conditions [17]. Therefore, in the diagnosis and treatment of septic lung injury, it is not enough to stay at the traditional cell type level. Positive interventions may not exert equal regulatory effects on one type of cell but rather have different regulatory potencies on different subpopulations of this type of cell.

In the current study, we validated the role of Maresin1 in inhibiting neutrophil infiltration in septic lung injury and then focused on the subpopulation bias of Maresin1 regulation of neutrophils. We classified neutrophils in septic lung injury into four subpopulations based on distinct molecular signatures. We found that sepsis altered the proportion of different subsets of neutrophils in lung tissue, while Maresin1 intervention had a more pronounced effect on the proportion of the Neutrophil3-Cxcl3 subpopulation. We performed a DEGs analysis and functional analysis of the Neutrophil3-Cxcl3 subpopulation and found that the functions of DEGs in the Neutrophil3-Cxcl3 subpopulation mainly focus on cytokine receptor binding, regulation of cytokine production, regulation of cell–cell adhesion, and promotion of neutrophils, monocytes, and other immune cells chemotaxis and recruitment. The genes most involved in the functional enrichment analysis were Cxcl3, Ccl3, Ccl4, Icam1, and Il1rn. All of the Cxcl3, Ccl3, Ccl4, and Icam1 play an important role in the migration of neutrophils [38,39,40,41]. It is worth noting that the expression of Il1rn, as an anti-inflammatory gene [42], was up-regulated in the CLP group, which may be caused by auto-protective regulation. Maresin1 can significantly reduce the expression of the Cxcl3, Ccl3, Ccl4, and Icam1 genes in the Neutrophil3-Cxcl3 subpopulation, which may be an important mechanism by which Maresin1 regulates neutrophil function, and also provides a possible intervention target for the treatment of septic lung injury.

Relb, Irf5, and Junb are important transcription factors driving neutrophil effector responses, while Nfkb2, Rfx2, and Relb promote neutrophil survival [43,44,45]. In addition to contributing to the expression and production of cytokines and chemokines, and various effector functions, Irf5, Junb, and Relb are also involved in regulating the transition from the blood into the tissue. Interestingly, the Neutrophil3-Cxcl3 subpopulation just specifically highly expressed Icam1 and a series of chemokines, such as Cxcl3, Ccl3, Ccl4, and so on. We deduced that the inflammatory conditions could lead to massive migration and infiltration of the Neutrophil3-Cxcl3 subpopulation into lung tissue. The recruitment of the Neutrophil3-Cxcl3 subpopulation will lead to the migration of other inflammatory cells to the lung tissue through the production of a series of cytokines and chemokines, forming an excessive inflammatory response, while Maresin1 can effectively inhibit this process.

There are also certain limitations in our current study. Firstly, although we revealed that the regulatory effect of Maresin1 on neutrophils is subpopulation biased, our current study mainly explored the effect of Maresin1 on the transcriptional level of neutrophils. For the Neutrophil-Cxcl3 subpopulation, we validated the presence of neutrophils that highly express the Cxcl3 gene by using public datasets but we did not validate whether Cxcl3 is also highly expressed at the protein level. We have not yet explored the exact role of the Neutrophil-Cxcl3 subpopulation. Exploring the important role of the Neutrophil-Cxcl3 subpopulation requires new and substantial efforts, including siRNA infection, construction of Cxcl3 knockout animals, etc. Elucidating the role and mechanism of the Neutrophil-Cxcl3 subpopulation is the focus of our next study. Secondly, our present study focused on the subpopulation on which Maresin1 has a major effect. While other interesting findings, such as the high expression of the keratin gene Krt83 in some neutrophils, were not explored. We will continue to explore this in depth in the future. In addition, another noteworthy problem is that Maresin1 is very unstable and needs to be stored at −80 °C. This problem seriously limits the application of Maresin1, which needs more exploration to overcome.

In conclusion, we confirmed that the regulatory effect of Maresin1 on neutrophils during septic lung injury is subpopulation biased. Maresin1 can effectively inhibit the infiltration of the Neutrophil3-Cxcl3 subpopulation and inhibit the expression of key inflammatory genes and transcription factors in the Neutrophil3-Cxcl3 subpopulation. Our studies suggested that the exploration of sepsis lung injury should not only stop at the level of certain types of cells but also explore the heterogeneity of cells in depth, which is conducive to discovering the pathological mechanism of the disease and new treatment strategies.

## Figures and Tables

**Figure 1 cells-11-03733-f001:**
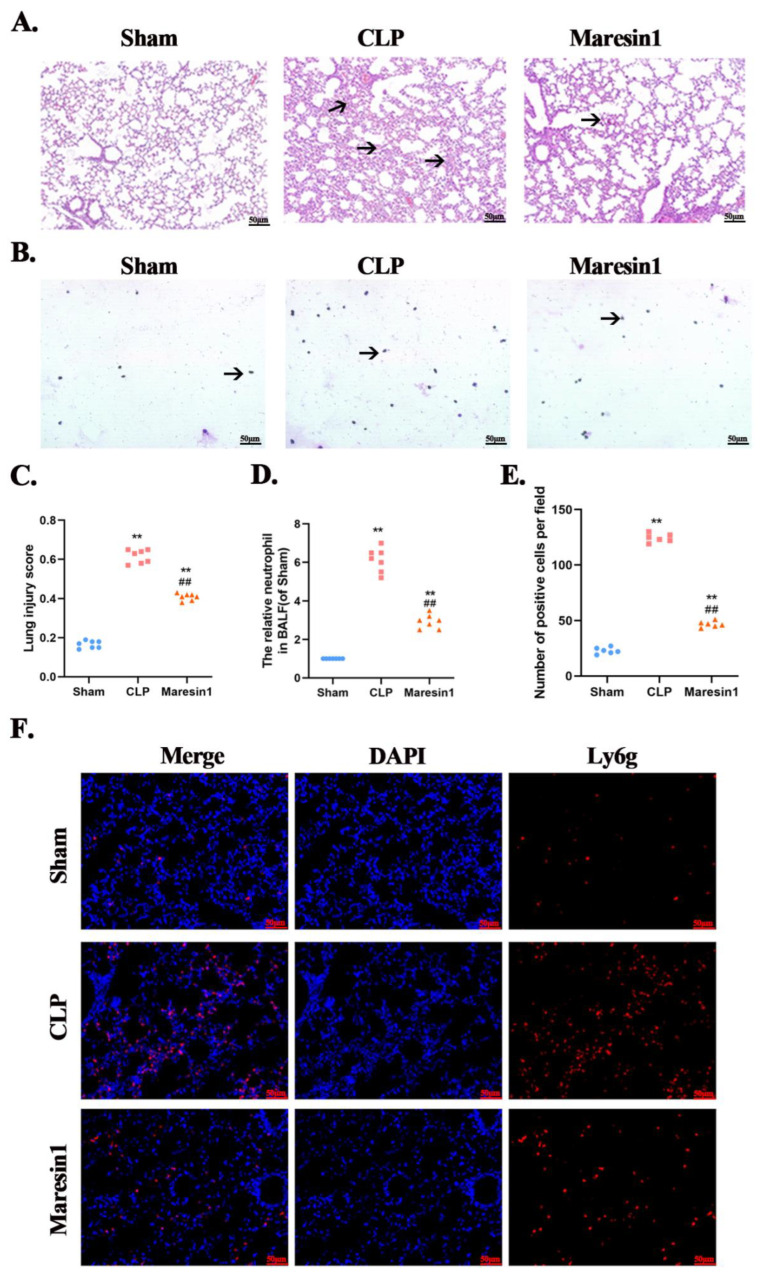
Maresin1 attenuates lung injury and neutrophil infiltration in septic lung injury. (**A**): The representative lung tissue HE staining results of different groups of mice (scale: 50 μm, magnification 200×). (**B**): The number of neutrophils in the BALF was counted by using the Wright–Giemsa method. (**C**): The lung injury scores of all groups. (**D**): The relative proportions of neutrophils in different groups of BALF counted by the Wright–Giemsa method. the results of the neutrophil count in BALF. (**E**): The statistical results of neutrophil immunofluorescence staining corresponding to (**F**). (**F**): The immunofluorescence staining results of the neutrophils in the lung tissues e (scale: 50 μm, magnification 200×). Data were presented as means ± SEM. ** *p* < 0.01 versus the Sham group; ## *p* < 0.01 versus the CLP group.

**Figure 2 cells-11-03733-f002:**
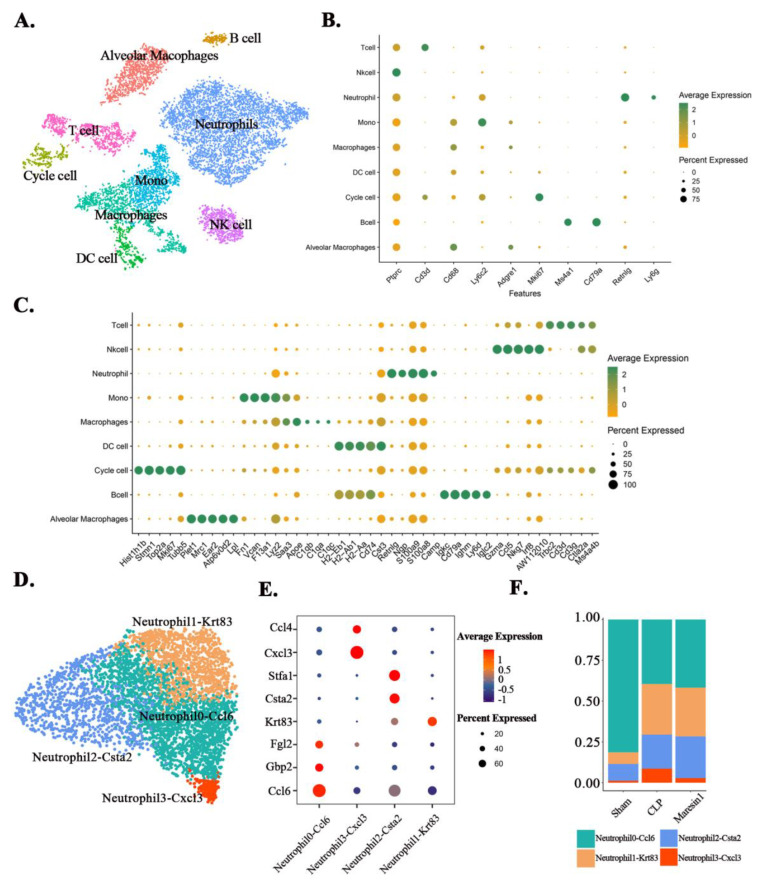
Single-cell sequencing revealed the cellular component of the neutrophil populations. (**A**): The tsne plot of all immune cells (Cd45+), showing a total of 9 distinct immune cell types that were identified. The different cell types are colored as indicated on the label. (**B**): The dot plot of marker genes for different types of immune cells. (**C**): The top5 DEGs for different immune cell clusters. (**D**): The umap plot of the cellular composition of neutrophils in lung tissue. A total of four clusters of neutrophils were identified. (**E**): The dot plot for the principal marker genes of different neutrophil subpopulations. (**F**): The fraction of different neutrophil subpopulations.

**Figure 3 cells-11-03733-f003:**
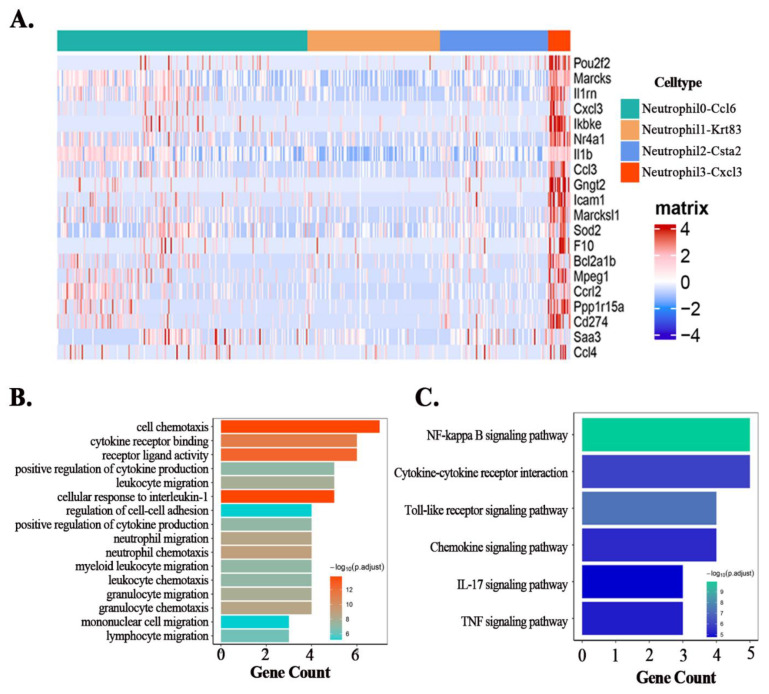
The DEGs and functional analysis of Neutrophil-cxcl3 subpopulation. (**A**): The heat map of the expression levels of the top20 DEGs of the Neutrophil-cxcl3 subpopulation. (**B**): The Go enrichment analysis of the DEGs of the Neutrophil-cxcl3 subpopulation. (**C**): The KEGG enrichment analysis of the DEGs of Neutrophil-cxcl3 subpopulation.

**Figure 4 cells-11-03733-f004:**
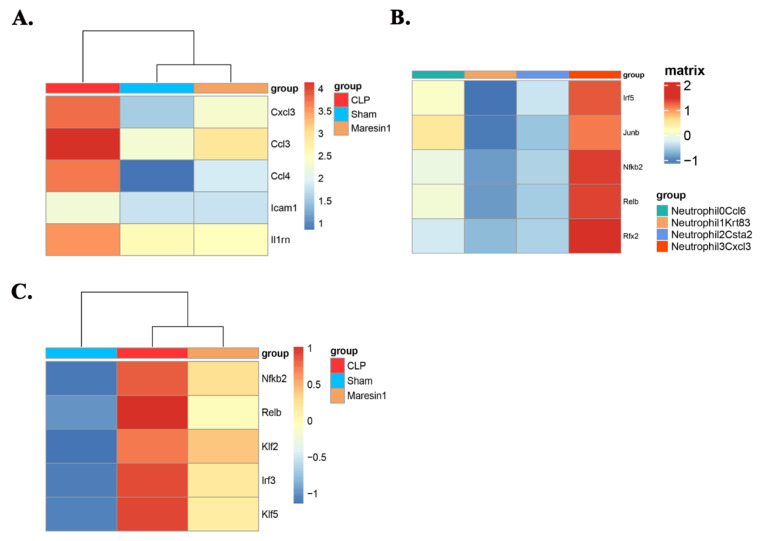
Maresin1 regulates gene expression and transcription factor expression of the neutrophil-Cxcl3 subset. (**A**): The heat map of the average expression levels of related genes for different groups. (**B**): The heat map of the regulon activity scores of TF motifs estimated per cell by pSCENIC. Shown are five differential activated motifs in the Neutrophil-cxcl3 subpopulation. (**C**): Maresin1 suppresses the expression levels of key transcription factors of the Neutrophil-cxcl3 subpopulation.

## Data Availability

Data and materials can be obtained by contacting the corresponding author. Raw data will be uploaded to a public database before publication.

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
