# Peer review of "Single-Cell Sequencing Reveals the Regulatory Role of Maresin1 on Neutrophils during Septic Lung Injury"

_cells, 2022, doi:10.3390/cells11233733_

Round 1

Reviewer 1 Report (Previous Reviewer 2)

The resubmitted version has made considerable modifications and addressed most of the concerns. It’s ready for acceptance and publication after raising the below issue. 

Fig 1B, the neutrophils staining in the BALF indicated by the arrow is hard to classify and identify.  It’s better to provide high-magnification pictures for each group.

Author Response

Dear reviewer:     

Thank you for your kind letter and for your constructive comments concerning our article entitled “Single-cell sequencing reveals the regulatory role of Maresin1 on neutrophils during septic lung injury” (Manuscript ID cells-1909258). According to your valuable comments, we have tried our best to modify our manuscript to meet the requirements within the allotted time. If you have any questions, please contact us without hesitation. Thank you again for your valuable suggestions to improve the quality of our manuscript.

Response

  1. The resubmitted version has made considerable modifications and addressed most of the concerns. It’s ready for acceptance and publication after raising the below issue. Fig 1B, the neutrophils staining in the BALF indicated by the arrow is hard to classify and identify.  It’s better to provide high-magnification pictures for each group.

Response: We have improved the image of Figure 1B so that the arrows can be better recognized. The changes are shown in Figure1B.

Kind regards,
Shanglong Yao
E-mail: yaoshanglong@hust.edu.cn

Reviewer 2 Report (Previous Reviewer 3)

Although the authors told me they would show the figure with the IF controls (answer my question 4), they did not. At least in the file that was sent to me I didn't find this figure.

Author Response

Dear reviewer:     

Thank you for your kind letter and for your constructive comments concerning our article entitled “Single-cell sequencing reveals the regulatory role of Maresin1 on neutrophils during septic lung injury” (Manuscript ID cells-1909258). These comments are all valuable and helpful for improving our article. According to your valuable comments, we have tried our best to modify our manuscript to meet the requirements within the allotted time. If you have any questions, please contact us without hesitation. Thank you again for your valuable suggestions to improve the quality of our manuscript.

Response

  1. Although the authors told me they would show the figure with the IF controls (answer my question 4), they did not. At least in the file that was sent to me I didn't find this figure.

Response: Thank you for your inquiry. We have provided IF control figures in the previous modification. Because of the system, the relevant Figures are not displayed in the web system. Please download the word attachment to view. Thank you again for your suggestions.

In this experiment, we used FITC and CY3-labeled secondary antibodies for immunofluorescence staining, We used these two secondary antibodies alone for control, and the results showed that under the same excitation wavelength conditions, the secondary antibodies had no significant effect on the fluorescence observation results. The results are shown in the figure below.

Kind regards,
Shanglong Yao
E-mail: yaoshanglong@hust.edu.cn

Reviewer 3 Report (New Reviewer)

The authors disclosed the molecular mechanism of a previously identified anti-inflammatory pro-resolving eicosanoid class of lipid using single cell sequencing technology. For basic research purpose and in order to understand the putative MoA, this is sure a great attempt. However, translational ability of Marsein-1 to an anti-septic drug is far away considering the physicochemical property of the molecule.  Marsein-1 is a very unstable compound with many conjugated double bonds and supposed to be stored at -80oC. No discussion on formulation, stability of the drug is discussed as this is important and to be considered in the beginning of the project. The authors published many papers on this molecule already, so should have thought about it.

The study design in CLP model to understand the activity of Marsein-1 on neutrophil  is not appropriate. CLP induced Sepsis and sepsis induced ALI is a sequential process, polymicrobial infection induced systemic inflammation and then lung and other organ injury happens >24-48h. Not all peritonitis/polymicrobial infection leads to ALI in 24h. Please explain why you choose this study design. Only 1 dose is studied, is that good enough for treating a sepsis patient? 

Some of the fundamental concept is not clearly presented. Such as if Marsein-1 polarizes macrophages to M2 phenotype and anti-inflammatory by nature and decreases the neutrophil activity then how this is beneficial for sepsis in clearing pathogens? Neutrophils help in clearing bacteria and microbes. Sepsis is not only inflammation but also infection and microbial load. How this current data from single cell experiment sheds light on this issue? In clinical practice antibiotics will be used to treat sepsis and lung injury, the study should include a group with antibiotics and antibiotics+Marsein-1 to establish the synergy.

Suggestion to use sepsis induced acute lung injury is advised.

 Please provide the catalog no. for the antibodies used.

The bar graphs in Fig. 1C-E should show all dots from individual measurement to represent robust and rigor data.

The fluoroscent intensity bar graph of the IHC (Fig. 1F) is missing. What is the sample size, how many times the experiments are repeated?

Author Response

Dear reviewer:     

Firstly, we would like to thank you for your kind letter and for your constructive comments concerning our article entitled “Single-cell sequencing reveals the regulatory role of Maresin1 on neutrophils during septic lung injury” (Manuscript ID cells-1909258). These comments are all valuable and helpful for improving our article. According to your valuable comments, we have tried our best to modify our manuscript to meet the requirements within the allotted time. In this revised version, changes to our manuscript within the document were all highlighted by using red-colored text. Point-by-point responses are listed below in this letter. If you have any questions, please contact us without hesitation. Thank you again for your valuable suggestions to improve the quality of our manuscript.

Response

  1. The authors disclosed the molecular mechanism of a previously identified anti-inflammatory pro-resolving eicosanoid class of lipid using single cell sequencing technology. For basic research purpose and in order to understand the putative MoA, this is sure a great attempt. However, translational ability of Marsein-1 to an anti-septic drug is far away considering the physicochemical property of the molecule.  Marsein-1 is a very unstable compound with many conjugated double bonds and supposed to be stored at -80oC. No discussion on formulation, stability of the drug is discussed as this is important and to be considered in the beginning of the project. The authors published many papers on this molecule already, so should have thought about it. The study design in CLP model to understand the activity of Marsein-1 on neutrophil  is not appropriate. CLP induced Sepsis and sepsis induced ALI is a sequential process, polymicrobial infection induced systemic inflammation and then lung and other organ injury happens >24-48h. Not all peritonitis/polymicrobial infection leads to ALI in 24h. Please explain why you choose this study design. Only 1 dose is studied, is that good enough for treating a sepsis patient?  Some of the fundamental concept is not clearly presented. Such as if Marsein-1 polarizes macrophages to M2 phenotype and anti-inflammatory by nature and decreases the neutrophil activity then how this is beneficial for sepsis in clearing pathogens? Neutrophils help in clearing bacteria and microbes. Sepsis is not only inflammation but also infection and microbial load. How this current data from single cell experiment sheds light on this issue? In clinical practice antibiotics will be used to treat sepsis and lung injury, the study should include a group with antibiotics and antibiotics+Marsein-1 to establish the synergy. Suggestion to use sepsis induced acute lung injury is advised.

Response: Thank you very much for your constructive comments. In this study, we used the CLP model to simulate sepsis, which is currently the most widely recognized model to study septic lung injury(1, 2). In addition, we performed a pathological examination on the lung tissues of all mice to confirm the occurrence of lung injury. Therefore, we thought the CLP model is appropriate for our research purposes. In the current study, we demonstrated that the intervention of Maresin1 could significantly improve septic lung injury and reduce neutrophil infiltration. Our previous studies have shown that the positive effect of Maresin1 on sepsis is dose-dependent(3), so we did not set up the concentration gradient of Maresin1 in this study. As you said above, Maresin1 is indeed unstable, so it needs to be stored at - 80 ℃. Solving this problem is of great significance to promote its application, but it requires more researchers to participate. We added the related discussion in the Discussion section.

One of the important reasons why the mechanism of sepsis has not been clarified is the complexity of the immune environment. As you said, neutrophils are an important weapon for the body to resist external stimuli, but excessive inflammation caused by excessive neutrophils will damage the tissue. It is of positive significance to inhibit excessive neutrophil recruitment in the early stage of sepsis-induced lung injury. Our current study mainly suggested that Maresin1 can significantly reduce neutrophil infiltration in septic lung injury, and this regulatory effect has a distinct subpopulation bias. In addition to reducing the proportion of the Neutrophil-Cxcl3 subpopulation in the neutrophil, maresin1 could inhibit the expression of related inflammatory genes and key transcription factors in the Neutrophil-Cxcl3 subpopulation. In the introduction, we mentioned the effect of Maresin1 on the polarization of macrophages, just to show that Maresin1 has a broad regulatory effect on immune cells. However, the regulatory effect of Maresin1 on immune cells is subpopulation biased, which should be paid more attention to.

    More and more in-depth studies are needed to explore the pathogenesis and effective treatment strategies of sepsis. Maresin1 may be an effective tool for sepsis treatment, but more verification is needed. At present, our main purpose is to explore the subgroup bias of Maresin1 on the regulation of neutrophils in septic lung injury. Therefore, we did not combine other antibiotics. We thought that fewer variables can achieve our goal more directly. We will continue to carry out research on Maresin1, and we also expect that Maresin1 combined with antibiotics can better improve the organ damage caused by sepsis. Thanks again for your constructive comments.

  1. Please provide the catalog no. for the antibodies used.

Response: Thank you for your inquiry. We have supplemented the relevant antibody information in the manuscript. The changes are shown in red.

3.The bar graphs in Fig. 1C-E should show all dots from individual measurement to represent robust and rigor data.The fluoroscent intensity bar graph of the IHC (Fig. 1F) is missing.

    Response: Thank you very much for your constructive suggestions. We have modified Figure 1. We use scatter plot instead of the histogram, and supplement the statistical graph of immunofluorescence results. The changes are shown in Figure 1B

4.What is the sample size, how many times the experiments are repeated?

 Response: Thank you for your inquiry. All tests have been repeated more than six times. We have added relevant details in the manuscript. The changes are shown in red.

Kind regards,
Shanglong Yao
E-mail: yaoshanglong@hust.edu.cn

References

  1. Jiao Y, Zhang T, Zhang C, Ji H, Tong X, Xia R, et al. Exosomal miR-30d-5p of neutrophils induces M1 macrophage polarization and primes macrophage pyroptosis in sepsis-related acute lung injury. Critical care (London, England). 2021;25(1):356.
  2. Wang JF, Wang YP, Xie J, Zhao ZZ, Gupta S, Guo Y, et al. Upregulated PD-L1 delays human neutrophil apoptosis and promotes lung injury in an experimental mouse model of sepsis. Blood. 2021;138(9):806-10.
  3. Wang F, Wang M, Wang J, Chen M, Sun S, Yao S, et al. Maresin1 ameliorates sepsis-associated lung injury by inhibiting the activation of the JAK2/STAT3 and MAPK/ NF-κB signaling pathways. Microbial pathogenesis. 2020;148:104468.

Round 2

Reviewer 3 Report (New Reviewer)

Authors have addressed most of the concerns

This manuscript is a resubmission of an earlier submission. The following is a list of the peer review reports and author responses from that submission.

Round 1

Reviewer 1 Report

In this manuscript, by performing the sc-RNA seq, the authors tried to clarify the regulatory effect of Maresin1 in CLP-induced lung injury. The authors found that Maresin1 can significantly reduce Neutrophil-Cxcl3 subpopulation and inhibit the expression of related inflammatory genes and key transcription factors in the Neutrophil-Cxcl3 subpopulation, which the authors think could be the potential mechanism by which Maresin1 regulates septic lung injury.

The Major concern is that in this manuscript, no direct evidence showed that Maresin1 regulated septic lung injury by altering the Neutrophil-cxcl3 population. The author cited other datasets to prove the presence of Neutrophil-cxcl3 population, but no data or literature prove the importance of this population. The author performed DEG analysis, and found some important cytokines and TFs which are known to play a key role in neutrophil functions, however, these are only hypotheses, and direct experiments and evidence need to be performed and shown to validate the role of  neutrophil-cxcl3 in septic lung injury and the alteration of this population is the potential mechanism by which Maresin1 regulates septic lung injury.

minor points:

1. Figure 1A is blurred due to the low resolution. The authors claimed that the lung of the CLP group had disordered lung tissue structure, thickened alveolar septa, accumulation of inflammatory cells in the alveoli, and the pulmonary septum, and bleeding and protein debris exudation in the alveolar cavity. It is better to use some markers to point these out in the picture to make the readers easy to understand.

2. Scale was lacking in Figure 1 E.

Author Response

Response to Reviewer1

1.The Major concern is that in this manuscript, no direct evidence showed that Maresin1 regulated septic lung injury by altering the Neutrophil-cxcl3 population. The author cited other datasets to prove the presence of Neutrophil-cxcl3 population, but no data or literature prove the importance of this population. The author performed DEG analysis, and found some important cytokines and TFs which are known to play a key role in neutrophil functions, however, these are only hypotheses, and direct experiments and evidence need to be performed and shown to validate the role of  neutrophil-cxcl3 in septic lung injury and the alteration of this population is the potential mechanism by which Maresin1 regulates septic lung injury

     Response: Thank you very much for your comments. Our current study mainly showed that Maresin1 can significantly reduce neutrophil infiltration in septic lung injury, and this regulatory effect has a distinct subpopulation bias. The regulatory effect of Maresin1 on neutrophils is more concentrated in the Neutrophil-Cxcl3 subpopulation. As you mentioned above, our findings are mainly based on the results of single-cell sequencing that Maresin1 can significantly reduce the infiltration of the Neutrophil-Cxcl3 subpopulation and inhibit the expression of related inflammatory genes and key transcription factors in the Neutrophil-Cxcl3 subpopulation. We found that there is also an obvious Neutrophil-Cxcl3 subpopulation in the lung injury-related data published by other researchers, which suggests that the Neutrophil-Cxcl3 subpopulation may have important significance in the occurrence and development of the disease. Exploring the important role of the Neutrophil-Cxcl3 subpopulation requires new and substantial efforts, including siRNA infection, construction of Cxcl3 knockout animals, etc. We have added additions to the limitations of the current work in the discussion section and this is also the focus of our next research, thank you again for your comments.

  1. Figure 1A is blurred due to the low resolution. The authors claimed that the lung of the CLP group had disordered lung tissue structure, thickened alveolar septa, accumulation of inflammatory cells in the alveoli, and the pulmonary septum, and bleeding and protein debris exudation in the alveolar cavity. It is better to use some markers to point these out in the picture to make the readers easy to understand.

Response: Thank you very much for your careful suggestions. We have adjusted Figure1 A to make it clearer. We have also added some arrow marks to point out relevant indications for lung injury scores, including disordered lung tissue structure, thickened alveolar septa, accumulation of inflammatory cells in the alveoli, etc. in Figure1 A to make the readers easy to understand.

  1. Scale was lacking in Figure 1 E.

Response: Thank you very much for your careful suggestions, we have supplemented the relevant content in the Legend of Figure1 E. The changes are shown in red in the manuscript.

Reviewer 2 Report

This study found that Maresin1 has regulatory effects on neutrophils during a septic lung inflammatory process. Moreover, this regulatory effect of Maresin1 mainly focused on the Neutrophil-Cxcl3 subpopulation. This is a wonderful and interesting research paper and provides a new perspective on the regulation of neutrophils function during septic lung injury. However, there are some concerns that need to be further addressed and improved.

Major concerns:

1.       Abstract should be refined and reorganized. For eg. Our study suggested that Maresin1 can significantly reduce neutrophil infiltration ……….. of Maresin1 on neutrophils is more concentrated in the Neutrophil-Cxcl3 subpopulation. These two sentences can be combined.

2.       Fig 1.  1) authors should mark the neutrophils in the HE stains with a representative label; 2) There missed scale bar in A and E;  3)  authors have collected the BALF and it’s better to show the neutrophils in each group by using Giemsa staining.

3.       According to the results, there seems to be a supplementation of some validation data, and this is important for increasing the readability. The first required validation is that Maresin1 on neutrophils is more concentrated in the Neutrophil-Cxcl3 subpopulation. Another required validation is that the decreased inflammation of Maresin1-challenged mice is associated with neutrophil3-Cxcl3 subpopulation dynamics.

Minor concerns:

1.       MI, USA; Cat.10878) in 100ul of normal saline, please change the “ul” into “µL” ?

2.       Fig 1 legend, P maybe “P”.

3.       In Fig 1, whether the panel A and E made from the serial sections? If so, it looks like they cannot match.

Author Response

Response to Reviewer2

This study found that Maresin1 has regulatory effects on neutrophils during a septic lung inflammatory process. Moreover, this regulatory effect of Maresin1 mainly focused on the Neutrophil-Cxcl3 subpopulation. This is a wonderful and interesting research paper and provides a new perspective on the regulation of neutrophils function during septic lung injury. However, there are some concerns that need to be further addressed and improved. 

1.Abstract should be refined and reorganized. For eg. Our study suggested that Maresin1 can significantly reduce neutrophil infiltration ……….. of Maresin1 on neutrophils is more concentrated in the Neutrophil-Cxcl3 subpopulation. These two sentences can be combined.

Response: Thank you very much for your constructive comments. We have revised the abstract section. The changes are shown in red in the manuscript.

2.Fig 1.  1) authors should mark the neutrophils in the HE stains with a representative label; 2) There missed scale bar in A and E;  3)  authors have collected the BALF and it’s better to show the neutrophils in each group by using Giemsa staining.

Response: Thank you very much for your careful suggestions. We have adjusted Figure1 A to make it clearer. We have also added some arrow marks to point out relevant indications for lung injury scores, including disordered lung tissue structure, thickened alveolar septa, accumulation of inflammatory cells in the alveoli, etc. in Figure1 A to make the readers easy to understand. We have supplemented the relevant scale bar in the Legend of Figure1 A,E. We counted neutrophils after staining BALF with Giemsa, and the statistical results were not significantly different from the current results.

3.According to the results, there seems to be a supplementation of some validation data, and this is important for increasing the readability. The first required validation is that Maresin1 on neutrophils is more concentrated in the Neutrophil-Cxcl3 subpopulation. Another required validation is that the decreased inflammation of Maresin1-challenged mice is associated with neutrophil3-Cxcl3 subpopulation dynamics.

Response: Thank you very much for your comments. Our current study mainly showed that Maresin1 can significantly reduce neutrophil infiltration in septic lung injury, and this regulatory effect has a distinct subpopulation bias. Our findings are mainly based on the results of single-cell sequencing that Maresin1 can significantly reduce the infiltration of the Neutrophil-Cxcl3 subpopulation and inhibit the expression of related inflammatory genes and key transcription factors in the Neutrophil-Cxcl3 subpopulation. We found that there is also an obvious Neutrophil-Cxcl3 subpopulation in the lung injury-related data published by other researchers, which suggests that the Neutrophil-Cxcl3 subpopulation may have important significance in the occurrence and development of the disease. Exploring the important role of the Neutrophil-Cxcl3 subpopulation requires new and substantial efforts, including siRNA infection, construction of Cxcl3 knockout animals, etc. We have added additions to the limitations of the current work in the discussion section and this is also the focus of our next research, thank you again for your comments.

Reviewer 3 Report

cells-1909258

-The procedures used for euthanasia of animals should be described in detail (anesthetic, final procedures of death, disposal of carcasses and etc).

-The time period for sample collection in mice is not presented in the text of MS. Including just the citation of previous articles for this information makes it very difficult to read. The materials and methods section should be reformulated in all subsections, granting to the readers most of possible information regarding the procedures.

Major points:

-It is well established in the literature that in sepsis, the migratory/activation capacity (microbicidal activity/NETs etc.) of neutrophils is drastically impaired; therefore, the authors need to explain better and in a bruising manner, how would reducing neutrophil activity (even in a specific subpopulation) in a clinical condition in which there is already impairment of this function be beneficial? In fact, robust data from the literature suggest that the reestablishment of the function of these leukocytes is crucial for the treatment of sepsis. How do the authors explain this paradoxical result/thought they defend? Please include this discussion in the text.

-Are the neutrophils in the lung microcirculation (adhered/aggregated in the vascular endothelium) or in the extravascular lung tissue? The HE images, although do not identify leukocytes in the interalveolar space, do not allow a deeper analysis, as the magnification used in the images, and even the images themselves, are not adequate. Double IF with blood vessel markers (e.g. neutrophil markers + CD31, ICAM-1/CD54) will adress this important question.

The perfusion of animals before sample collection may also represent an interesting procedure for this question, as it would eliminate blood components in the IF analysis. This seems to me to be a fundamental point to be evaluated in the present study.

-For the authors' hypothesis to be viable, additional mechanistic investigations are necessary; in vivo experiments using anti-cxcl3 antibodies to observe the evolution of sepsis (the reviewer noted that there is already data in this sense about its receptor; as well as for cxcl3 in vitro, but apparently not yet in vivo), as well as measuring cxcl3 protein and mRNA levels in lung tissue in CLP and CLP+mar1 animals are necessary.

-To what extent is Cxcl3 expressed in neutrophils, which could be sufficient to modify the course of sepsis-induced ALI? This is not clear from the discussion.....other cells such as macrophages and basal cells of the respiratory epithelium, are described as a major cell expressing cxcl3 in humans. Thus, would inhibiting a cell type that is not described (at least for now) as the main cell expressing Cxcl3 really be enough to modify the course of sepsis/ALI? Experiments suggested above may address these points. Additionally, how to translate this rationally to the human considering the broad spectrum of action of SPMs on the immune system?

Author Response

Response to Reviewer3

1.The procedures used for euthanasia of animals should be described in detail (anesthetic, final procedures of death, disposal of carcasses and etc).

Response: Thank you very much for your constructive comments. We have revised the animals chapter of the Materials and methods section. We supplemented the anesthesia, handling procedures, and sacrifice of experimental animals. The changes are shown in red in the manuscript.

2.The time period for sample collection in mice is not presented in the text of MS. Including just the citation of previous articles for this information makes it very difficult to read. The materials and methods section should be reformulated in all subsections, granting to the readers most of possible information regarding the procedures.

Response: Thank you very much for your suggestion. Our experimental specimen collection time point is 24 hours after surgery, and we have supplemented and revised the relevant content in the materials and methods section. The changes are shown in red in the manuscript.

3.It is well established in the literature that in sepsis, the migratory/activation capacity (microbicidal activity/NETs etc.) of neutrophils is drastically impaired; therefore, the authors need to explain better and in a bruising manner, how would reducing neutrophil activity (even in a specific subpopulation) in a clinical condition in which there is already impairment of this function be beneficial? In fact, robust data from the literature suggest that the reestablishment of the function of these leukocytes is crucial for the treatment of sepsis. How do the authors explain this paradoxical result/thought they defend? Please include this discussion in the text.

Response: Thank you very much for your inquiry. The essence of sepsis is multiple organ dysfunction caused by the dysregulation of the body's immune response to infection, including an initial phase of hyperinflammatory response followed by a phase of immunosuppression. The time point of our experiment was 24h of sepsis. As shown in Figure 1E, at this stage, the infiltration of neutrophils in the lung tissue increased significantly. How to inhibit the excessive inflammatory response at this stage is of great significance to the lung injury caused by sepsis. We supplement relevant content and literature in the discussion section.

4.Are the neutrophils in the lung microcirculation (adhered/aggregated in the vascular endothelium) or in the extravascular lung tissue? The HE images, although do not identify leukocytes in the interalveolar space, do not allow a deeper analysis, as the magnification used in the images, and even the images themselves, are not adequate. Double IF with blood vessel markers (e.g. neutrophil markers + CD31, ICAM-1/CD54) will adress this important question. The perfusion of animals before sample collection may also represent an interesting procedure for this question, as it would eliminate blood components in the IF analysis. This seems to me to be a fundamental point to be evaluated in the present study

Response: Thank you very much for your constructive suggestions. We performed double-labeled fluorescent staining on the lung tissues of different groups of mice. Due to the limitation of current antibodies, we selected Icam1 and the neutrophil marker Ly6g for preliminary observation. The results are shown in Supplementary Figure 1. Compared with the Sham group, the infiltration of neutrophils in the lung tissue of the mice in the CLP group was significantly increased. Due to the destruction of the lung tissue structure caused by sepsis, the neutrophils infiltrated diffusely, and the intervention of Maresin1 could significantly improve this phenomenon.

5.For the authors' hypothesis to be viable, additional mechanistic investigations are necessary; in vivo experiments using anti-cxcl3 antibodies to observe the evolution of sepsis (the reviewer noted that there is already data in this sense about its receptor; as well as for cxcl3 in vitro, but apparently not yet in vivo), as well as measuring cxcl3 protein and mRNA levels in lung tissue in CLP and CLP+mar1 animals are necessary.

Response: Thank you very much for your constructive advice. Our current study mainly reveals that the regulation of neutrophils in septic lung injury by Maresin1 is subpopulation-biased, focusing mainly on the Neutrophil-Cxcl3 subpopulation. Neutrophil-Cxcl3 subpopulation is a subpopulation of neutrophils with high expression of Cxcl3, but Cxcl3 is not only expressed in neutrophils. The use of the Cxcl3 antibody not only reduces the expression of Cxcl3 in neutrophils and does not demonstrate the precise mechanism of the Neutrophil-Cxcl3 subpopulation in septic lung injury. We have also found subpopulations of neutrophils with high expression of Cxcl3 in single-cell sequencing data from other lung injury studies, so we prefer to use conditional knockout Cxcl3 mice and then sort neutrophils in lung tissue to compare with neutrophils from wild-type mice to determine the exact role of the Neutrophil-Cxcl3 subpopulation in septic lung injury, which is what our future work will explore. In the current study, we found the subpopulation bias of Maresin1's regulatory role for neutrophils in septic lung injury, but the role of the Neutrophil-Cxcl3 subpopulation has not been explored in depth. We believe that our findings still have implications for the mechanism and treatment of septic lung injury, and we also describe the limitations of our study and future work in the discussion section.

6.To what extent is Cxcl3 expressed in neutrophils, which could be sufficient to modify the course of sepsis-induced ALI? This is not clear from the discussion.....other cells such as macrophages and basal cells of the respiratory epithelium, are described as a major cell expressing cxcl3 in humans. Thus, would inhibiting a cell type that is not described (at least for now) as the main cell expressing Cxcl3 really be enough to modify the course of sepsis/ALI? Experiments suggested above may address these points. Additionally, how to translate this rationally to the human considering the broad spectrum of action of SPMs on the immune system?

    Response: Thank you very much for your comments. Our current study mainly showed that Maresin1 can significantly reduce neutrophil infiltration in septic lung injury, and this regulatory effect has a distinct subpopulation bias. The regulatory effect of Maresin1 on neutrophils is more concentrated in the Neutrophil-Cxcl3 subpopulation. We cannot guarantee that targeting the Neutrophil-Cxcl3 subpopulation can alter the course of septic ALI, but we found that Maresin1 intervention can significantly reduce the infiltration of the Neutrophil-Cxcl3 subpopulation and inhibit the expression of related inflammatory genes and key transcription factors in the Neutrophil-Cxcl3 subpopulation and thereby ameliorate lung injury. Our findings suggested that we should focus on immune cell subsets in the treatment of septic lung injury. We consider our work to be suggestive for the exploration of the pathogenesis and treatment of septic lung injury.

Round 2

Reviewer 1 Report

The current version is good enough for publishing.

Author Response

Response to Reviewer1

1.The current version is good enough for publishing. 

Response: Thank you very much for your constructive review and acknowledgment of our studies, which inspired our further exploration of immune cell heterogeneity in septic lung injury. Thanks again!

Reviewer 2 Report

The resubmitted manuscript has made some modifications and raised concerns. However, there are still some required experiments that should be addressed.

1.       The authors have collected the BALF and it’s better to show the neutrophils in each group by using Giemsa staining, this is important data for evidence of Maresin1 on neutrophils function

2.       According to the results and conclusion, there seems to lack strong evidence of Maresin1 on neutrophils is more concentrated in the Neutrophil-Cxcl3 subpopulation, which especially needs validation in mice septic lung injury model of this single-cell sequence data.

3.       It’s better to evidence that the decreased inflammation of Maresin1-challenged mice is associated with neutrophil3-Cxcl3 subpopulation dynamics.

Author Response

Dear reviewer:     

    Firstly, we would like to thank you for your kind letter and for your constructive comments concerning our article entitled “Single-cell sequencing reveals the regulatory role of Maresin1 on neutrophils during septic lung injury” (Manuscript ID cells-1909258). These comments are all valuable and helpful for improving our article. According to your valuable comments, we have tried our best to modify our manuscript to meet the requirements within the allotted time. In this revised version, changes to our manuscript within the document were all highlighted by using red-colored text. Point-by-point responses are listed below in this letter. Our results mainly rely on single-cell sequencing, and there is indeed a lack of some in-depth exploration, such as the use of gene knockout animals. We will continue to further explore the immune cell heterogeneity in septic lung injury in the future. Nonetheless, we consider our work to be suggestive for the exploration of the pathogenesis and treatment of septic lung injury. If you have any questions, please contact us without hesitation. Thank you again for your valuable suggestions to improve the quality of our manuscript.

Response

1.The authors have collected the BALF and it’s better to show the neutrophils in each group by using Giemsa staining, this is important data for evidence of Maresin1 on neutrophils function.

Response: Thank you very much for your constructive suggestions. We counted neutrophils after staining BALF with Giemsa and the results are shown in Figure 1B.

2.According to the results and conclusion, there seems to lack strong evidence of Maresin1 on neutrophils is more concentrated in the Neutrophil-Cxcl3 subpopulation, which especially needs validation in mice septic lung injury model of this single-cell sequence data.

Response: Thank you very much for your comments. Our current study mainly showed that Maresin1 can significantly reduce neutrophil infiltration in septic lung injury, and this regulatory effect has a distinct subpopulation bias. As you stated above, Our findings are mainly based on the results of single-cell sequencing, which is a tool for single-cell studies with a high degree of accuracy and specificity. Our results suggested that Maresin1 can significantly reduce the proportion of the Neutrophil-Cxcl3 subpopulation and inhibit the expression of related inflammatory genes and key transcription factors ,which have been shown to play an important role in neutrophils, in the Neutrophil-Cxcl3 subpopulation. We also found that there is also an obvious Neutrophil-Cxcl3 subpopulation in the lung injury-related data published by other researchers, which suggests that the Neutrophil-Cxcl3 subpopulation may have important significance in the occurrence and development of the disease. Based on the above results, we consider that we should be able to draw the conclusion that the regulation of Maresin1 on neutrophils is subpopulation bias. We sincerely hope for your understanding and thank you again for your comments.

3.It’s better to evidence that the decreased inflammation of Maresin1-challenged mice is associated with neutrophil3-Cxcl3 subpopulation dynamics.

Response: Thank you very much for your comments. We firstly demonstrated that the intervention of Maresin1 could significantly reduce neutrophil infiltration in septic lung injury by immunofluorescence staining, cell counting, etc. Then we analyzed all the neutrophils at the transcriptional level using single-cell sequencing. Our current studies suggested that the regulation of neutrophils in septic lung injury by Maresin1 is subpopulation-biased. In addition to reducing the proportion of the Neutrophil-Cxcl3 subpopulation in the neutrophil, maresin1 could inhibit the expression of related inflammatory genes and key transcription factors in the Neutrophil-Cxcl3 subpopulation. Taking these results together, we consider our current study to be credible. We strongly agree with other valid approaches, such as the use of Cxcl3 knockout mice, to further confirm our conclusions. But with our current study, we believe that our findings still have implications for the mechanism and treatment of septic lung injury, and we also describe the limitations of our study in the discussion section. We would very much appreciate your understanding. Thanks again for your comments.

Kind regards,
Shanglong Yao
E-mail: yaoshanglong@hust.edu.cn

Reviewer 3 Report

Minor points:

Introduction section:

(subitem 2.2) - Sacrifice does not sound good as a scientific term; euthanasia seems more suitable.

(subitem 2.5) - The number of samples per animal and slides per group should be included; additionally, data regarding the antibody (Icam1) used in the new experiment is absent; authors should provide specifications of products used. 

-The thickness of the sample sections; and how many fields were considered per sample during the analysis should be mentioned;

-The authors performed analyzes using slides with secondary antibodies alone in parallel as controls to ensure that unspecified staining did not occur? This issue must be adressed.

-Scale bars are missing in Fig. 1A and E, and in Supp Fig. 1, and should be provided.

Conclusion section:

-The reviewer understands as essential that the authors describe in the conclusion, that although the findings are important for the advancement of the understanding of the immunopathogenesis of septic ALI, they do not guarantee that such an approach modifies the outcome of the disease.

Author Response

Dear reviewer:     

    Firstly, we would like to thank you for your kind letter and for your constructive comments concerning our article entitled “Single-cell sequencing reveals the regulatory role of Maresin1 on neutrophils during septic lung injury” (Manuscript ID cells-1909258). These comments are all valuable and helpful for improving our article. According to your valuable comments, we have tried our best to modify our manuscript to meet the requirements within the allotted time. In this revised version, changes to our manuscript within the document were all highlighted by using red-colored text. Point-by-point responses are listed below in this letter. Thank you again for your valuable suggestions to improve the quality of our manuscript. If you have any questions, please contact us without hesitation.

Response

1.(subitem 2.2) - Sacrifice does not sound good as a scientific term; euthanasia seems more suitable.

Response: Thank you very much for your careful suggestions. We have changed " sacrificed " to "euthanized". The change is shown in red in the manuscript.

2.(subitem 2.5) - The number of samples per animal and slides per group should be included; additionally, data regarding the antibody (Icam1) used in the new experiment is absent; authors should provide specifications of products used. 

Response: Thank you very much for your careful suggestions. We used 6 specimens per group for fluorescent staining. ICAM1 antibody was purchased from Thermo Fisher Scientific. We have supplemented the relevant instructions in the manuscript. The changes are shown in red in the manuscript.

3.The thickness of the sample sections; and how many fields were considered per sample during the analysis should be mentioned;

Response: Thank you very much for your inquiry. The thickness of the sample sections was 4 μm, and we observed 5 fields for each sample, and took the average value of the number of positive cells for statistics. We have supplemented the relevant instructions in the manuscript. The changes are shown in red in the manuscript.

4.The authors performed analyzes using slides with secondary antibodies alone in parallel as controls to ensure that unspecified staining did not occur? This issue must be adressed.

Response: Thank you very much for your inquiry. In this experiment, we used FITC and CY3-labeled secondary antibodies for immunofluorescence staining, We used these two secondary antibodies alone for control, and the results showed that under the same excitation wavelength conditions, the secondary antibodies had no significant effect on the fluorescence observation results. The results are shown in the figure in the attachment.

5.Scale bars are missing in Fig. 1A and E, and in Supp Fig. 1, and should be provided.

Response: Thank you very much for your careful suggestions. We have supplemented the relevant scale bar Figure1 A,E and supplementary Figure1.

6.The reviewer understands as essential that the authors describe in the conclusion, that although the findings are important for the advancement of the understanding of the immunopathogenesis of septic ALI, they do not guarantee that such an approach modifies the outcome of the disease.

Response: Thank you very much for your understanding and acknowledgment of our studies, which inspired our further exploration of immune cell heterogeneity in septic lung injury. Thanks again!

 Kind regards,
Shanglong Yao
E-mail: yaoshanglong@hust.edu.cn
